# SelM: Selective Mechanism based Audio-Visual Segmentation

## ABSTRACT

Audio-Visual Segmentation (AVS) aims to segment sound-producing objects in videos according to associated audio cues, where both modalities are affected by noise to different extents, such as the blending of background noises in audio or the presence of distracted objects in video. Most existing methods focus on learning interactions between modalities at high semantic levels but is incapable of filtering low-level noise or achieving fine-grained representational interactions during the early feature extraction phase. Consequently, they struggle with illusion issues, where nonexistent audio cues are erroneously linked to visual objects. In this paper, we present SelM, a novel architecture that leverages selective mechanisms to counteract these illusions. SelM employs State Space model for noise reduction and robust feature selection. By imposing additional bidirectional constraints on audio and visual embeddings, it is able to precisely identify crutial features corresponding to sound-emitting targets. To fill the existing gap in early fusion within AVS, SelM introduces a dual alignment mechanism specifically engineered to facilitate intricate spatio-temporal interactions between audio and visual streams, achieving more fine-grained representations. Moreover, we develop a cross-level decoder for layered reasoning, significantly enhancing segmentation precision by exploring the complex relationships between audio and visual information. SelM achieves state-of-the-art performance in AVS tasks, especially in the challenging Audio-Visual Semantic Segmentation subset. Source code will be made publicly available.

## CCS CONCEPTS

• **Computing methodologies** → **Video segmentation**.

## KEYWORDS

Audio-visual segmentation, selective mechanism, multimodal feature alignment

## 1 INTRODUCTION

Audio-Visual Segmentation (AVS) is an emerging multimedia technology with wide applications, such as video editing, industrial maintenance, and surveillance. It aims at leveraging audio cues to enable models to locate sound-producing objects and generate pixel-level segmentation masks. Introduced by AVSBench [56, 57], this task encompasses three settings: Semi-supervised Single Sound Source Segmentation (S4), Fully-supervised Multiple Sound Source

*MM '24, 28 October - 1 November 2024, Melbourne, Australia*
© 2024 Copyright held by the owner/author(s). Publication rights licensed to ACM.
ACM ISBN 978-1-4503-XXXX-X/18/06
https://doi.org/XXXXXXX.XXXXXXX

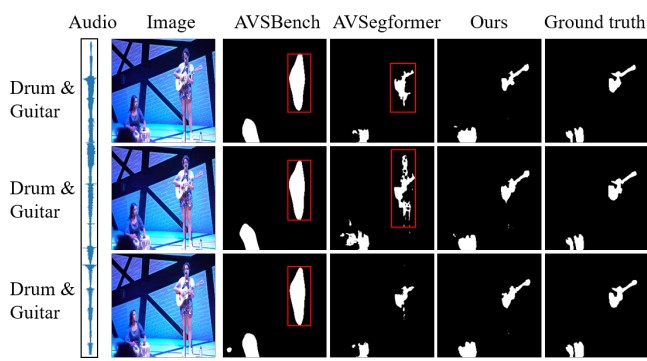

**Figure 1: Auditory Illusions. Illustrating the struggle of existing methods with auditory illusion issues, where models erroneously segment non-emitting targets.**

Segmentation (MS3), and Audio-Visual Semantic Segmentation (AVSS), covering a spectrum of segmentation complexities.

In the AVS task, video provides explicit spatial information, while audio reveals changes in sound over time. For instance, if the model detects human vocals suddenly emerging within a piano accompaniment through the audio, then the model should accordingly segment both the singer and the piano. Similarly, upon observing guitar strings being plucked in the video, it should accordingly segment the guitar. Therefore, the comprehensive integration of both modalities is crucial for accurately inferring the location of entities. Current methods [8, 29, 32, 38, 57] predominantly focus on developing more effective auditory-visual information fusion methods. For example, AVSBench [57] integrates the TPVAI module to facilitate interaction between Audio-Vidual features, while CATR [29] introduces a decoupled Audio-Visual Transformer for integration across both spatial and temporal dimensions. From another perspective, some researches [18, 39, 57] explores the diversity of feature representation. AVSBench employs regularization to reduce the distance between modal features, narrowing the domain gap. One representative study is ECMVAE [39], which utilizes multiple regularizations to decompose features into shared and specific components, achieving more robust feature representations.

Despite significant advancements in quantitative evaluation metrics achieved by current methods, extensive visualizations unveil 'hallucination' as depicted in Figure 1. Despite the performers do not sing, current methods are influenced to different degrees and erroneously segment the silent objects. We term this phenomenon the *Auditory Illusions* issue, highlighting that in existing methods, audio fails to play a decisive role in entity selection. We identify two reasons: (1) Both audio and video contain considerable noise, and when multiple sound sources appear in the same scene, key information overlaps, challenging the model's ability to decouple different sound-emitting entities, thereby misleading its segmentation. (2) Although current methods employ feature fusion or regularization, they predominantly utilize intermediate or late fusion techniques,

as depicted in Figure 2. Given the substantial disparities between the modalities, previous approaches result in a deficiency of nuanced fusion and fine-grained representations.

To address the first issue, we consider leveraging the selective mechanism of Mamba [11] to suppress noise. Mamba, renowned for its powerful long-sequence modeling capability and computational efficiency, garners widespread attention within the community. Its reliable modeling prowess stems from the selective mechanism that compresses redundant representations and noise, isolating key and robust features. This capability is particularly suited to the AVS task, prompting our use of Mamba coupled with the design of a bidirectional conditional constraint for further cross-modal cue selection. For the second issue, beyond facilitating high-level information exchange, we also propose the dual alignment approach between the two encoders to achieve early fusion. The differences between the previous method and ours are demonstrated in Figure 2(a) and (b). Numerous multimodal studies [7, 10, 42, 54] highlight the benefits of early fusion, such as the capability to overcome domain gaps and generate more detailed representations. Thus, for the first time in the context of AVS tasks, we introduce a dual alignment module as an early fusion strategy to enhance the information diversity and uniformity of the feature distributions.

Specifically, similar to AVSBench [57], we utilize two encoders to extract features from video and audio. Between these encoders, we design a Dual Alignment Module (DAM) for early fusion, facilitating fine-grained interaction. The encoded features are then directed to a Bidirectional Conditioned Selective Mechanism Module (BCSM) for the selection of relevant information and the imposition of bidirectional constraints. To ensure a comprehensive understanding during the decoding phase, we devise a Cross-LEVEl Reasoning (CLEVER) decoder that alternately performs high-level fusion and decoding of Audio-visual information, which yields the segmentation mask. Lastly, the auxiliary loss is applied to the cross-attention map of the decoder to further refine segmentation quality.

In summary, our contributions are threefold:

- We employ a selective mechanism to filter out noise from features, using bidirectional constraints to select spatio-temporal information relevant to the sound-emitting entities, addressing the issue of *Auditory Illusions*.
- Transcending traditional intermediate fusion approaches, we propose an elegant alignment module dedicated to early fusion, cultivating more nuanced and enriched representations. This innovation fills a crucial gap in the current fusion methodologies for AVS tasks.
- We introduce an cross-level reasoning decoder that alternately merges and analyzes information from both modalities, yielding precise segmentation results.

## 2  RELATED WORK

### 2.1  Audio-Visual Segmentation

The purpose of Sound Source Localization (SSL) is to identify objects within a video that produce sound. Early efforts [1, 2, 5, 23, 24, 24, 33, 44, 44, 45, 47] in this domain remain at the patch level due to the lack of fine-grained annotations. Recent research [57] introduces the Audio-Visual Segmentation benchmark with pixel-level annotations, capturing considerable attention from communities.

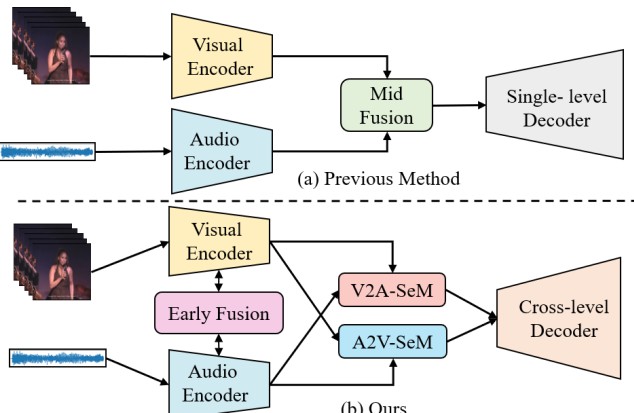

Figure 2: Compared with existing methods. (a) Existing methods typically facilitate feature interaction at high level, often employing intermediate fusion with single-level decoder. (b) Our approach introduces early fusion, utilizes a bidirectional selection mechanism module at intermediate stages, and interlaced in the final stages.

A considerable portion of the latest studies [18, 31, 39] focuses on effectively integrating audio-visual information. For example, AVS-Bench [57] proposes the TPAVI module for multimodal information integration, incorporating regularization terms into optimization objectives to bridge the gap between audio and visual features. AVS-BG [18] utilizes a bidirectional generation manner to reinforce the audio-visual relationship through maintaining cycle consistency. ECMVAE [39] divides the features of the two modalities into shared and specific knowledge to enhance representation diversity. CATR [29] introduces a decoupled audio-visual transformer, taking audio features and learnable embeddings as queries. AVSC [31] employs a two-stage paradigm consisting of instance segmentation followed by sound source classification to alleviate ambiguity. Our research aims to address the problem of datasets bias-induced **Auditory Illusions**, effectively leveraging audio information and preventing model degradation.

### 2.2  State Space Models

The State Space models, originating from classical control theory [27], is introduced into the field of deep learning to model dependencies in long sequences. Recent advancements in the State Space model are driven by a variety of works [12–15, 17, 19, 35, 40, 48] that focus on refining its computational efficiency and analytical robustness. Notably, there is a particular focus on Mamba [11], which introduces a selective mechanism to enhance the model's information compression capabilities, while leveraging hardware-aware algorithms to accelerate computation. Due to its input-dependent parameters, Mamba demonstrates formidable capabilities in modeling long sequences and possesses linear scalability properties. A wide array of research [16, 25, 34, 43, 58] harnesses Mamba for the extraction of hierarchical features in images, addressing directional sensitivity by augmenting the model with varied scanning directions. Mamba even shows promising performance in point

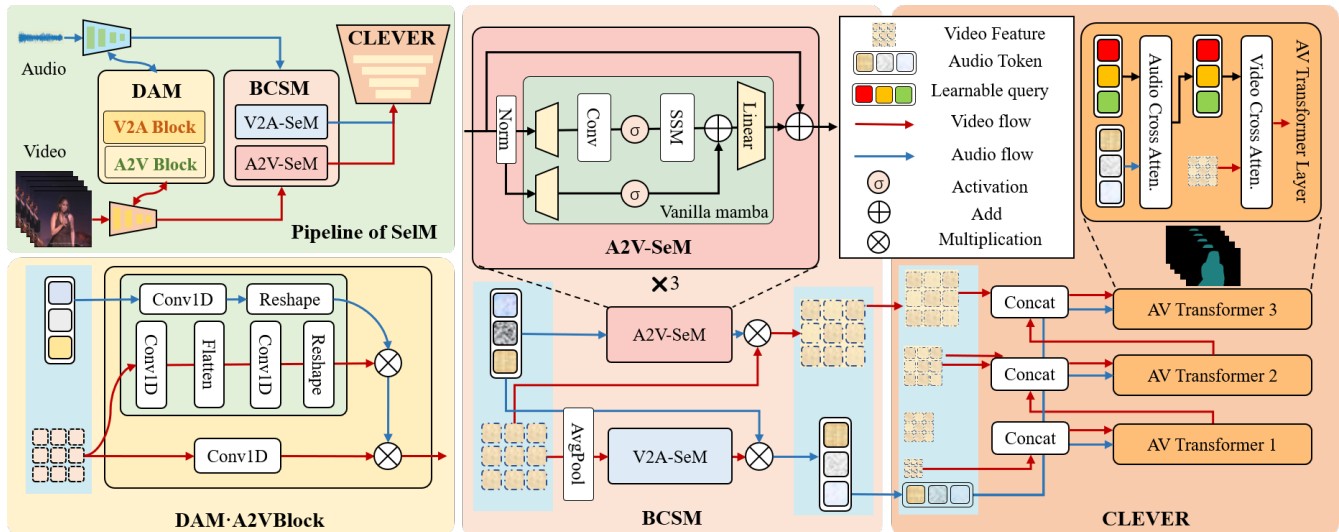

(a) DAM: Dual Alignment Module    (b) BCSM: Bidirectional Conditioned Selective Module    (b) CLEVER: Cross LEVEl Reasoning Decoder

**Figure 3: The pipeline of the proposed SelM model is depicted in the top-left part, featuring three novel design components. (a) During the encoding phase, a dual alignment strategy is employed to align the distributions of the two modalities. (b) After alignment, a selective mechanism is applied for noise suppression and robust feature representation, with bidirectional constraints yielding information pertinent to the sound-emitting objects. (c) In the decoding phase, learnable queries alternately interrogate both modalities, culminating in the predicted segmentation results.**

cloud analysis [30] and generative tasks [4, 25]. Audio-Visual Segmentation (AVS) depends crucially on the careful selection and highlighting of visual segments that match specific audio cues, ensuring segmentation targets only relevant data from both modalities. Consequently, delving into Mamba's selective mechanism for AVS tasks presents an intriguing idea. To our knowledge, this marks the first attempt to refine Mamba's selection mechanism for AVS tasks, indicating a new direction in this field of study.

## 2.3 Staged Fusion Methods in Multimodal Learning

Integrating features and aligning latent representations across modalities are key for multimodal tasks [3, 7, 10, 26, 42, 50, 52, 54]. Early fusion, the process of combining modalities before processing, leverages all available data to effectively capture intermodal correlations. Building on this foundation, intermediate fusion takes place at a model's mid-stage, where it merges features to enhance the understanding of complex data relationships. Finally, late fusion, applied at the decision level, ensures modality independence by integrating insights only after individual analyses, thereby maintaining the distinctiveness of each data source. In the AVS field, both intermediate and late fusion techniques are widely used. TPAVI [57] leverages intermediate fusion to enhance audio-visual interactions, while AVSegFormer [8] applies late fusion, integrating audio-visual tokens in its decision stage to capitalize on the strengths of each modality. This research builds upon existing concepts while innovatively integrating early fusion to bridge the domain gap between auditory and visual pre-training. By doing so, it aims to achieve comprehensive multimodal integration and understanding at every stage.

## 3 PRELIMINARIES: SELECTIVE MECHANISM

State Space Models (SSMs) [19, 35, 40, 48] are usually employed as linear time-invariant systems to transform the input $\boldsymbol{x}(t) \in \mathbb{R}^M$ to the output $\boldsymbol{y}(t) \in \mathbb{R}^M$ via a hidden state $\boldsymbol{h}(t) \in \mathbb{R}^N$. Mathematically, this system can be formulated as a linear ordinary differential equation:

$$\begin{aligned} \boldsymbol{h}'(t) &= \boldsymbol{A}\boldsymbol{h}(t) + \boldsymbol{B}\boldsymbol{x}(t), \\ \boldsymbol{y}(t) &= \boldsymbol{C}\boldsymbol{h}(t) + D\boldsymbol{x}(t), \end{aligned} \tag{1}$$

where $\boldsymbol{A} \in \mathbb{R}^{N \times N}$ is the state transition matrix, $\boldsymbol{B} \in \mathbb{R}^{N \times 1}$ and $\boldsymbol{C} \in \mathbb{R}^{1 \times N}$ are projection matrices, and $D \in \mathbb{R}^1$ denotes the skip connection. Considering that the continuous-time models cannot be directly applied to deep learning algorithms, the structured state space sequence models and Mamba [11] employ the principle of zero-order hold (ZOH) for discretization, resulting in the following formula:

$$\begin{aligned} \overline{\boldsymbol{A}} &= exp\left(\triangle \boldsymbol{A}\right), \\ \overline{\boldsymbol{B}} &= \left(\triangle \boldsymbol{A}\right)^{-1} \cdot \left(exp\left(\triangle \boldsymbol{A}\right) - \boldsymbol{I}\right) \cdot \triangle \boldsymbol{B}, \end{aligned} \tag{2}$$

where the continuous parameters $A$ and $B$ are discretized into $\overline{A}$ and $\overline{B}$ through a time scale parameter $\triangle$. After discretization, Eq. (2) can be written as follows:

$$\begin{aligned} h_t &= \overline{\boldsymbol{A}}h_{t-1} + \overline{\boldsymbol{B}}x_t, \\ y_t &= \boldsymbol{C}h_t. \end{aligned} \tag{3}$$

The final output $y(t)$ can be computed through a convolution, directly establishing a relationship with the input $x(t)$. The formula is as follows:

$$\overline{K} = (C\overline{B}, C\overline{AB}, ..., C\overline{A}^{M-1}\overline{B}),$$
$$y(t) = x(t) * \overline{K},$$

(4)

where $\overline{K} \in \mathbb{R}^N$ is a structured convolutional kernel. Based on this, Mamba [11] further introduces an input-dependent modeling approach, and the selective mechanism can be expressed as follows:

$$\overline{B} = s_B(x),$$
$$\overline{C} = s_C(x),$$
$$\Delta = \tau_A(Parameter + s_A(x)),$$

(5)

where $s_B(\cdot)$, $s_C(\cdot)$, and $s_A(\cdot)$ are linear projections. $\tau_A(\cdot)$ denotes the Softplus activation function.

## 4 METHOD

This section starts by introducing the overall architecture of the proposed Selective Mechanism-based AVS method (SelM), followed by elaborating its core components.

### 4.1 Overall Architecture

The proposed SelM is built upon the Selective Mechanism to select and align the representative features of visual and audio inputs. It mainly consists of two feature encoders, a Dual Alignment Module (DAM), a Bidirectional Conditioned Selective Mechanism Module (BCSM), and an Audio-visual Cross-LEVEl Reasoning decoder (CLEVER). As shown in Figure 3, given the video frames and corresponding audio clip, they are first fed into the visual encoder and audio encoder, respectively, to obtain the multi-stage features. Meanwhile, the DAM is applied to align and fuse the audio and visual features gradually for all the stages. Then, BCSM is designed to model the spatial and temporal information, and filter out the feature noise from audio and visual in a bidirectional manner. Finally, the filtered multi-modal features are fed into CLEVER to obtain the final segmentation result. To make a fair comparsion, we follow the prior methods [22, 57] by employing ResNet50 [20] or Pyramid Vision Transformer (PVT-v2) [51] as the visual encoder and VGGish [22] as the audio encoder. In the following, we will describe the core components in detail.

### 4.2 DAM: Dual Alignment Module for Early Fusion

Given the multi-stage audio and visual features extracted by the encoders, the Dual Alignment Module for Early Fusion (DAM) is designed for cross-modal feature alignment and integration. Different from the conventional works that only employ late fusion for audio and vision features at the high semantic level, we propose a dual alignment manner and apply it at multiple feature stages to achieve sufficient integration between the two modalities. While the two modalities of data have different distribution characteristics, they exhibit strong semantic correlations and complementarity. Therefore, DAM is designed to comprise two symmetric subblocks named

V2ABlock and A2VBlock, to enhance the relevant information from audio and vision features upon each other, respectively.

To simplify the description, we will focus on providing a detailed description of the A2VBlock. The structure of the V2ABlock is identical to the A2VBlock, with the only difference being the interchange of data modalities between audio and video. As shown in Figure 3 (a), given the audio tokens $T$ and the visual feature $F$ from the same stage of encoders, A2VBlock aims to enhance the representative information from the visual features according to the audio tokens. Therefore, we design to learn an alignment weight map $W$ upon the audio-visual features. Specifically, both the audio tokens and visual features are fed into convolution branches, respectively, and their outputs are element-wise multiplicated to produce the weight map, which is further utilized to re-weight the visual feature and output the enhanced version $F^a$. In our designs, Conv1D is adopted as the convolutional layer, followed by a GELU [21] activation function and a dropout [49] operation to introduce non-linearity and prevent overfitting.

To ensure the model achieves thorough understanding and alignment across all levels, DAM is applied at all the feature stages, allowing a seamless harmonization of features between the two branches. Compared with the existing AVS works, our method adopting the proposed DAM enjoys at least two benefits. First, it utilizes both low-level fine-grained and high-level semantic features, offering comprehensive knowledge about the segmentation target. Second, with the light-weight DAM, the visual and audio information are bilaterally and gradually enhanced and aligned, bringing more reliable representations of the target features.

### 4.3 BCSM: Bidirectional Conditioned Selective Mechanism

To investigate whether source signals significantly impact model decisions, we visualize the predicated segmentation maps of the existing methods, which show a *Auditory Illusions* issue. As demonstrated in Figure 1, models erroneously segment out the non-sounding performers. This indicates that models tend to take shortcuts by memorizing the segmented objects associated with specific scenes rather than fully utilizing audio information. The issue is challenging to settle. First, the input data from two modalities are complex and contain noise, making it challenging to decouple the key information, especially in multi-source scenarios where critical cues overlap. Second, video frames and audio signals change over time, and they struggle to perform mutual selection effectively.

Based on the above analyses, we introduce the Bidirectional Conditioned Selective Mechanism Module (BCSM). Details can be found in Figure 3(b). Specifically, we employ the Mamba [11] to suppress the irrelevant information of the given input. The design of Mamba incorporates a selection mechanism that emphasizes robust feature representation while filtering out noise and redundant information, thereby making it highly suitable for the AVS task. In addition, we design a bidirectional constraint that discerns relevant information across both modalities.

As shown in Figure 3 (b), the BCSM consists of two symmetric branches. Taking the upper branch as an example, the aligned video feature $F^a$ is transformed into sequences through average pooling, then processed through V2A-SeM which stacks three consecutive

**Figure 4: Comparison results in MS3 and AVSS settings. These findings demonstrate that SelM effectively localizes sound-emitting objects, mitigating auditory illusion challenges.**

Mamba blocks for filtering out irrelevant information. This produces high-level weight maps which contain appearance cues about sound-emitting objects for selecting audio tokens by element-wise multiplication. The process mirrors in the lower branch, with the note that audio tokens bypass pooling operation. For a Mamba block, it comprises a vanilla Mamba equipped with state space model principles (recommended in Section 3 and Mamba [11]), supplemented by residual connections and normalization operations. Overall, A2V-SeM and V2A-Sem are utilized for denoising within each modality, further refined by bidirectional constraints for feature selection. Ultimately, after processing through BCSM, we obtain the selected $F^s$ and $T^s$.

### 4.4 CLEVER: Audio-visual Cross-Level Reasoning Decoder

Existing approaches [31] utilize the FPN [28] decoder, a single-level decoding technique. They usually combine audio tokens and video features via a TPAVI [57] module or other feature fusion module to create a mixed audio-visual representation. The combined tensors are directly used as inputs to the decoder without any discrimination.

To facilitate a comprehensive understanding and interaction of audio-visual information at various levels, we design the Cross-LEVEl Reasoning (CLEVER) decoder, depicted in Figure 3 (c). CLEVER is structured around three identical Audio-Visual (AV) Transformer layers, where each layer sequentially processes through audio cross-attention followed by video cross-attention mechanisms. This layered operation marks our primary distinction from previous single-level decoders. While single-level decoders may utilize multi-scale features, they do not engage in alternate reasoning between different modalities.

Specifically, for an AV Transformer layer, the input consists of two parts: the selected audio tokens $T_i^s$ and the video features $F_i^s$. The outputs of the first two layers are concatenated with the subsequent $F_{i+1}^s$, forming mixed features that are fed into the next AV Transformer layer. Each AV transformer layer comprises two sequentially arranged cross-attention modules. For audio cross-attention, selected tokens $T^s$ serve as keys and values, while for video cross-attention, the selected features $F^s$ assumes these roles. In the same AV Transformer layer, learnable tokens are shared between two cross-attention modules as queries. Through alternate reasoning and comprehension, the queries are imbued with meaning related to the sound-producing entities. In the third layer, we perform a dot product between the learnable tokens of outputs and mixed video features to produce the final estimated masks.

### 4.5 Loss Objective

For the S4 and MS3 settings, we employ a synergistic blend of DICE [41] and BCE (Binary Cross Entropy) loss functions, further augmented by an auxiliary loss component. This integrated approach is formalized as follows:

$$\mathcal{L} = \mathcal{L}_{DICE} + \mathcal{L}_{BCE} + \mathcal{L}_{Aux}. \tag{6}$$

In this formulation, the DICE [41] and standard BCE losses are leveraged to quantify the similarity between the estimated masks produced by the model and the ground truth. The auxiliary loss introduces an additional constraint on the intermediate layers of the CLEVER decoder. Specifically, the auxiliary loss is defined by the following equation:

$$\mathcal{L}_{Aux} = \sum_{1}^{3} \alpha_i \ell_i, \tag{7}$$

where $\ell_i$ represents the DICE loss between the upsampled heat map of the $i^{th}$ layer and the ground truth. The term $\alpha_i$ denotes the associated weight for the $i^{th}$ layer. For our implementation, we empirically set the weights $\alpha_1$, $\alpha_2$, and $\alpha_3$ to 0.001, 0.01, and 0.1, respectively.

For the AVSS setting, we forego the application of DICE loss, opting instead to utilize the second and third components delineated in Equation 6.

## 5 EXPERIMENTS

### 5.1 Implementation Details

*5.1.1 Datasets.* We validate our ideas across three settings: single-sound and multi-sound under AVSBench-object [57], and AVSBench-Semantic [56]. AVSBench-Object targets audio-visual segmentation with two subjects — Semi-supervised Single-sound Source Segmentation (S4) with 4932 videos, and Fully-supervised Multi-sound Source Segmentation (MS3) with 424 videos — each offering pixel-level segmentation across 23 categories. The 2023 expansion, AVSBench-Semantic (AVSS), enhances AVSBench-Object with semantic labels and 10-second clips, broadening the dataset for audio-visual semantic segmentation research.

*5.1.2 Training Details.* We train our model on an NVIDIA Tesla A100 using PyTorch, employing the AdamW optimizer with a initial learning rate of $2 \times 10^{-5}$. The batch size is set at 4, with training durations of 40 epochs for the S4 setting, 100 epochs for the MS3 setting, and 30 epochs for the AVSS setting. Unless specifically mentioned, video frames are uniformly resized to $224 \times 224$. For the audio and visual encoders, we adopt the same configuration as used in previous works [29, 57].

*5.1.3 Evaluation Metrics.* We employ the Jaccard index $M_J$ and F-score $M_F$ as our primary evaluation metrics. The $M_J$ metric evaluates the segmentation accuracy by measuring the overlap between predicted and ground truth areas. Meanwhile, $M_F$ provides a combined measure of precision and recall, which is formulated as follows:

$$M_F = \frac{(1 + \beta^2) \times precision \times recall}{\beta^2 \times precision + recall} \tag{8}$$

where $\beta^2$ is set to 0.3 in our experiments, as in [57].

| Method | Backbone | AVSS | |
|---|---|---|---|
| | | $M_J \uparrow$ | $M_F \uparrow$ |
| 3DC [36] | ResNet-18 | 17.3 | 21.6 |
| AOT [53] | ResNet-50 | 25.4 | 31.0 |
| AVSBench [57] | ResNet-50 | 20.2 | 25.2 |
| CATR [29] | ResNet-50 | - | - |
| AVSegformer [8] | ResNet-50 | 24.9 | 29.3 |
| **SelM** | ResNet-50 | **31.9** | **37.2** |
| AVSBench [57] | PVT-v2 | 29.8 | 35.2 |
| CATR [29] | PVT-v2 | 32.8 | 38.5 |
| AVSegformer [8] | PVT-v2 | 36.7 | 42.0 |
| **SelM** | PVT-v2 | **41.3** | **46.9** |

**Table 1: Comparison with state-of-the-art methods on the AVSS setting.**

### 5.2 Comparison with SOTA Methods.

*5.2.1 Quantitative comparisons.* To ensure a fair comparison, consistent with AVSBench [57], we employ ImageNet-1K [46] pre-trained ResNet-50 [20] or PVT-v2 [51] as the backbone for extracting visual features, alongside AudioSet [9] pre-trained VGGish [22] for audio feature extraction. We compare AVS-related research and SOTA methods within three settings: Semi-supervised Single Sound Source Segmentation (S4), Fully-supervised Multiple Sound Source Segmentation (MS3), and Audio-Visual Semantic Segmentation (AVSS). The related body of work includes sound source localization (LVS [5] and MSSL [44]), video object segmentation (3DC [36] and SST [6]), and salient object detection (iGAN [37]), each sharing some degree of relevance with AVS tasks.

Table 1 showcases our performance on the AVSS setting. Notably, SelM outperforms related works (3DC [36] and AOT [53]) and existing methods (AVSBench [57], CATR [29]), comprehensively surpassing the current best method, AVSegFormer [8], regardless of whether ResNet-50 or PVT-v2 is used for video feature extraction. For both $M_J$ and $M_F$ metrics, our approach achieves approximately an 11% performance increase over the baseline model AVSBench [56]. Table 2 presents the performance of SelM on the S4 and MS3 settings, where we slightly outperform current methods (AVS-BG [18], AUTR [32], AVSC [31], CATR [29], DiffusionAVS [38], ECMVAE [39], AVSegFormer [8], etc.) using PVT-v2 as the feature extractor, and we alse realize comparable or slightly superior performance with ResNet-50. It is particularly noteworthy that the AVSS task, with its more complex and varied sound sources and greater number of categories, is the most challenging. This underscores the capability of SelM to discern sound source information and achieve superior segmentation results.

*5.2.2 Qualitative Comparisons.* To showcase SelM's effectiveness in mitigating *Auditory Illusions*, we display comparative visualization results from AVSBench [57], AVSegFormer [8], and SelM. Figure 4 presents examples from the MS3 and AVSS settings, which involve multiple sound sources. For additional visualizations related to the S4 setting, please refer to the supplementary materials. It can be observed that previous methods struggle with *Auditory Illusions*

| Method | Backbone | S4 | | MS3 | |
|---|---|---|---|---|---|
| | | $M_J \uparrow$ | $M_F \uparrow$ | $M_J \uparrow$ | $M_F \uparrow$ |
| LVS [5] | ResNet-18 | 37.9 | 51.0. | 29.5 | 33.0 |
| MSSL [44] | ResNet-18 | 44.9 | 66.3 | 26.1 | 36.3 |
| 3DC [36] | ResNet-152 | 57.1 | 75.9 | 36.9 | 50.3 |
| SST [6] | ResNet-101 | 66.3 | 80.1 | 42.6 | 57.2 |
| iGAN [37] | ResNet-50 | 61.6 | 77.8 | 42.9 | 54.4 |
| LGVT [55] | Swin-B | 74.9 | 87.3 | 40.7 | 59.3 |
| AVSBench [57] | ResNet-50 | 72.8 | 84.8 | 47.9 | 57.8 |
| AVS-BG [18] | ResNet-50 | 74.1 | 85.4 | 45.0 | 56.8 |
| AUTR [32] | ResNet-50 | 75.0 | 85.2 | 49.4 | 61.2 |
| AVSC [31] | ResNet-50 | **77.0** | 85.2 | 49.6 | 61.5 |
| CATR [29] | ResNet-50 | 74.8 | **86.6** | 52.8 | 65.3 |
| ECMVAE [39] | ResNet-50 | 76.3 | 86.5 | 48.7 | 60.7 |
| AVSegformer [8] | ResNet-50 | 76.5 | 85.9 | 49.5 | 62.8 |
| **SelM** | ResNet-50 | 76.6 | 86.2 | **54.5** | **65.6** |
| AVSBench [57] | PVT-v2 | 78.7 | 87.9 | 54.0 | 64.5 |
| AVS-BG [18] | PVT-v2 | 81.7 | 90.4 | 55.1 | 66.8 |
| AUTR [32] | PVT-v2 | 80.4 | 89.1 | 56.2 | 67.2 |
| AVSC [31] | PVT-v2 | 80.6 | 88.2 | 58.2 | 65.1 |
| CATR [29] | PVT-v2 | 81.4 | 89.6 | 59.0 | 70.0 |
| ECMVAE [39] | PVT-v2 | 81.7 | 90.1 | 57.8 | 70.8 |
| AVSegformer [8] | PVT-v2 | 82.1 | 89.9 | 58.4 | 69.3 |
| **SelM** | PVT-v2 | **83.5** | **91.2** | **60.3** | **71.3** |

**Table 2: Comparison with state-of-the-art methods on the S4 and MS3 settings.**

issues, whereas our approach demonstrates stronger sound source localization capabilities and avoids segmenting irrelevant objects. Moreover, our segmentation is more precise and refined. This is due to the implementation of information interaction measures at various stages, especially via BCSM, which effectively suppresses noise and isolates robust features, thereby improving the precision of localization.

## 5.3 Ablation Studies

In this part, we detail the contributions of each module, the efficacy of symmetric design, and the impact of loss functions. To achieve a representative analysis, we conduct ablation studies under single-source (S4) and multi-source (MS3) audio settings, with all experiments utilizing the PVT-v2 as the video encoder. For additional subsidiary ablation studies, such as the effects of different stages in early fusion, please refer to the supplementary materials.

*5.3.1 Effect of the Essential Components.* To dissect the impact of each component on the effectiveness of SelM, we conduct ablation studies by sequentially removing the three modules from the complete SelM architecture. For these experiments, the S4 and MS3 settings are chosen for representative analysis. Regarding DAM and BCSM, these modules are simply removed, while for the condition 'without CLEVER', the decoder from AVSegFormer [8] is used as a substitute because it similarly utilizes the query-based transformer structure and operates on a single level. Table 3 shows

| Method | S4 | | MS3 | |
|---|---|---|---|---|
| | $M_J \uparrow$ | $M_F \uparrow$ | $M_J \uparrow$ | $M_F \uparrow$ |
| w.o. BCSM | 81.5(-2.0) | 89.8(-1.4) | 56.2(-4.1) | 67.8(-3.5) |
| w.o. DAM | 81.2(-2.3) | 89.5(-1.7) | 57.5(-2.8) | 68.8(-2.5) |
| w.o. CLEVER | 81.1(-2.4) | 89.7(-1.5) | 57.7(-2.6) | 68.8(-2.5) |
| **SelM** | **83.5** | **91.2** | **60.3** | **71.3** |

**Table 3: Effect of the essential components. Replacing or omitting any of our modules results in a decrease in performance.**

| Method | S4 | | MS3 | |
|---|---|---|---|---|
| | $M_J \uparrow$ | $M_F \uparrow$ | $M_J \uparrow$ | $M_F \uparrow$ |
| Add | 81.2(-2.3) | 89.7(-1.5) | 54.7(-5.6) | 66.6(-4.7) |
| Concat | 64.8(-18.7) | 78.6(-12.6) | 42.6(-17.7) | 53.0(-18.3) |
| **DAM** | **83.5** | **91.2** | **60.3** | **71.3** |

**Table 4: Effects of early fusion strategies.**

that eliminating any module reduces the performance of SelM, with removal of BCSM having the most substantial effect under MS3 setting. This emphasizes the effectiveness of our design in enhancing the selection and interaction of information in complex scenes. Furthermore, the incorporation of DAM and CLEVER plays distinct roles in bolstering performance.

**Dual Alignment Module.** Additionally, to validate the necessity of introducing the Dual Alignment Module for early fusion, we compare concatenation or addition of the features $F$ and tokens $T$ output from the intermediate layers of both encoders. To align the dimensions, we apply average pooling to the $F$ and a straightforward repeat operation to the $T$. The results, displayed in Table 4, indicate that neither of these elementary approaches results in performance gains, and simple concatenation leads to significant decline in capability. This indicates that a thoughtful design of early fusion is necessary as merely interacting the modalities does not automatically enhance performance.

**Bidirectional Conditioned Selective Mechanism Module.** For BCSM, we employ a qualitative analysis approach, visualizing the video feature maps with and without BCSM processing. Comparative observations in Figure 5 reveal that without processing by the BCSM module, interference object such as the dog exhibits notably high responses, leading to segmentation failure. In contrast, maps tend to ignore the interfering, with attention directed towards sound-emitting objects. This indicates that selective mechanism is effective, as stacking Mamba blocks suppresses noise in both modalities, and through bidirectional constraints, the model enhances its focus on information relevant to sound-producing objects.

**Cross-LEVEl Reasoning Decoder.** To further analyze the benefits of CLEVER for the AVS task, we perform ablation studies by replacing the cross-level reasoning with single modality reasoning. Specifically, we maintain the same number of cross-attention operations and queries but restrict the focus to purely video features or audio tokens. As shown in Table 5, resorting to any single

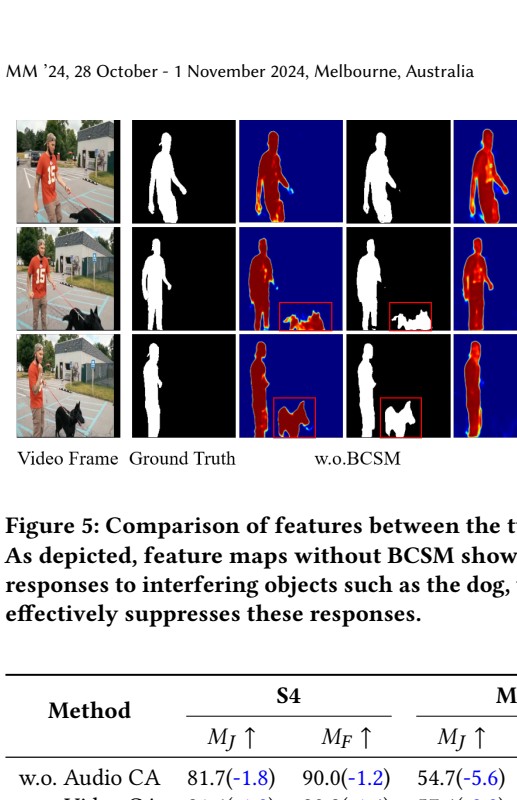

Video Frame    Ground Truth    w.o.BCSM    w.BCSM

**Figure 5: Comparison of features between the two versions. As depicted, feature maps without BCSM show substantial responses to interfering objects such as the dog, while BCSM effectively suppresses these responses.**

| Method | S4 | | MS3 | |
|---|---|---|---|---|
| | $M_J \uparrow$ | $M_F \uparrow$ | $M_J \uparrow$ | $M_F \uparrow$ |
| w.o. Audio CA | 81.7(-1.8) | 90.0(-1.2) | 54.7(-5.6) | 66.9(-4.4) |
| w.o. Video CA | 81.6(-1.9) | 89.8(-1.4) | 57.4(-2.9) | 67.8(-3.5) |
| **CLEVER** | **83.5** | **91.2** | **60.3** | **71.3** |

**Table 5: Ablation study on CLEVER. "CA" stands for Cross-Attention.**

| Method | S4 | | MS3 | |
|---|---|---|---|---|
| | $M_J \uparrow$ | $M_F \uparrow$ | $M_J \uparrow$ | $M_F \uparrow$ |
| A2V Block | 81.3(-2.2) | 89.7(-1.5) | 56.9(-3.4) | 66.6(-4.7) |
| V2A Block | 81.7(-1.8) | 90.2(-1.0) | 57.3(-3.0) | 67.6(-3.7) |
| **DAM** | **83.5** | **91.2** | **60.3** | **71.3** |

**Table 6: Impact of dual modality alignment in early fusion.**

modality results in performance degradation. Particularly, excluding audio interactions proves to be significantly detrimental in the MS3 setting. This is attributed to MS3 being a multi-source setting where a single video may contain multiple sound-emitting objects, rendering audio information crucial. This also demonstrates the capacity of CLEVER to adequately incorporate information from both modalities.

*5.3.2 Impact of the Symmetrical Pipeline.* To delve into the interplay between audio and visual modalities, as well as to quantify the benefits brought by symmetric design, we conduct ablation studies on the DAM and BCSM modules. In these experiments, "A2V" denotes the alignment or selection of video features based on audio information, producing video features as output, whereas "V2A" represents the reverse, yielding audio tokens as output. Table 6 illustrates the relationship between unilateral modeling capabilities and symmetric design during the early fusion with DAM. We

| Method | S4 | | MS3 | |
|---|---|---|---|---|
| | $M_J \uparrow$ | $M_F \uparrow$ | $M_J \uparrow$ | $M_F \uparrow$ |
| A2V-SeM | 81.1(-2.4) | 89.4(-1.8) | 56.7(-3.6) | 67.4(-3.9) |
| V2A-SeM | 81.3(-2.2) | 89.8(-1.4) | 57.1(-3.2) | 68.3(-3.0) |
| **BCSM** | **83.5** | **91.2** | **60.3** | **71.3** |

**Table 7: Ablation study on BCSM.**

| Method | S4 | | MS3 | |
|---|---|---|---|---|
| | $M_J \uparrow$ | $M_F \uparrow$ | $M_J \uparrow$ | $M_F \uparrow$ |
| BCE | 81.1(-2.4) | 89.4(-1.8) | 55.5(-4.8) | 65.9(-5.4) |
| +DICE | 81.3(-2.2) | 89.7(-1.5) | 56.7(-3.6) | 66.4(-4.9) |
| **+DICE+Aux** | **83.5** | **91.2** | **60.3** | **71.3** |

**Table 8: Ablation study on loss function**

find that V2A alignment is superior to A2V, and a symmetric alignment method exceeds any unilateral approach. Similarly, Table 7 reveals that selecting audio tokens using video features is more beneficial, and implementing bidirectional constraints yields the best performance.

*5.3.3 Impact of the loss functions.* We conduct ablation studies on the loss functions within the S4 and MS3 settings to thoroughly understand the contributions of each component. Table 8 demonstrates that incorporating DICE loss and auxiliary losses both contribute to enhance the final predictions. We observe that adding auxiliary losses leads to greater improvements in the MS3 setting, likely because the MS3 setting has less training data, and auxiliary losses help the model converge more effectively.

## 6 CONCLUSION

In this paper, we introduce the SelM model to address the *Auditory Illusions* issue in Audio-Visual Segmentation (AVS) tasks. We pioneer the application of early fusion in AVS by designing a Dual Alignment Module that enables fine-grained feature interactions and aligns the distributions of both modalities, filling the existing gap in early fusion approaches. We then incorporate the selective mechanism of Mamba to suppress noise and model robust representations, while bidirectional constrains further select relevant sound-emitting information. Finally, we develop a cross-level decoder that uses learnable queries to alternately interrogate audio and video cues, facilitating comprehensive interactions between the modalities and achieving effective segmentation results. Our approach sets new state-of-the-art performance across all three AVS settings and is thoroughly validated by extensive quantitative experiments. Qualitative visualizations confirm that SelM effectively mitigates *Auditory Illusions* issues.

**Future work:** The effectiveness of selective mechanism in AVS tasks is validated in this paper. We will fully utilize mamba as a feature extractor for both modalities to achieve more robust feature representations in the future.

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
