# OpenReview forum: "SelM: Selective Mechanism based Audio-Visual Segmentation"
_acmmm.org/ACMMM/2024/Conference — MM2024 Oral_

### Official Review · Reviewer_R9Uv · 2024-05-23

**Rating:** 3
**Confidence:** 2

**Summary:**

The paper introduces SelM, a novel architecture for Audio-Visual Segmentation (AVS). SelM addresses the limitations of existing methods that often fail to filter low-level noise or achieve fine-grained representational interactions during the early feature extraction phase. It employs a State Space model for noise reduction and robust feature selection, and a dual alignment mechanism for early fusion of audio and visual features. Additionally, a cross-level decoder is developed for layered reasoning, enhancing segmentation precision. The method is evaluated on the AVSBench dataset and demonstrates state-of-the-art performance in AVS tasks.

**Strengths:**

1. Novel Approach: The introduction of the selective mechanism and dual alignment module addresses the critical issue of auditory illusions and improves the precision of AVS tasks. This dual alignment strategy facilitates early fusion, which is a significant departure from traditional methods that primarily focus on high-level semantic interactions.
2. Technical Correctness: The use of a State Space model for noise reduction is technically sound and innovative. The paper effectively demonstrates how this model can suppress noise and isolate key features, which is crucial for accurate AVS.
3. Comprehensive Evaluation: The paper provides extensive experimental results, including ablation studies and comparisons with state-of-the-art methods. The results show significant improvements in segmentation accuracy and robustness across multiple settings.
4. Clarity and Presentation: The methodology is well-explained, and the use of figures and tables effectively illustrates the model architecture and experimental results. The qualitative comparisons further highlight the superiority of SelM over existing methods.
5. Applications: The proposed method has wide-ranging applications in multimedia, video editing, industrial maintenance, and surveillance, where precise AVS is critical.

**Limitations:**

1. Complexity: The introduction of multiple novel modules, including the dual alignment module and the cross-level reasoning decoder, adds to the complexity of the model. This might make it challenging to implement and optimize in practical scenarios.
2. Generalization: The evaluation is primarily conducted on the AVSBench dataset. Additional experiments on other datasets and real-world scenarios would strengthen the claims about the generalization capabilities of SelM.
3. Sensitivity to Parameters: The performance of the model is sensitive to the choice of certain hyperparameters, such as the weights in the loss function. This might require careful tuning and could impact the ease of use in different applications.
4. Fine-Grained Feature Extraction: While the selective mechanism helps in filtering out noise, there may still be challenges in distinguishing between very similar sound sources or visual objects, as noted in some qualitative analyses.

**Suitability:**

3

---

### Official Review · Reviewer_7WBu · 2024-05-24

**Rating:** 5
**Confidence:** 2

**Summary:**

The paper introduces the SelM model for Audio-Visual Segmentation (AVS), utilizing a novel early fusion strategy and a Dual Alignment Module to address Auditory Illusions. It incorporates noise suppression mechanisms and a cross-level decoder to enhance interaction between audio and video cues, significantly advancing AVS performance.

**Strengths:**

This paper focuses on noise reduction and robust feature selection by applying state space models to audio-visual segmentation (AVS), which is a novel approach. Additionally, the paper is clear and well organized.

**Limitations:**

I have some minor questions regarding the paper.

[1] Could you please provide the input and output dimensions for each layer in the CLEVER module?

[2] In Table 7, the performance appears lower when using A2V-SeM or V2A-SeM compared to not using BCSM at all in Table 3. Could you explain the reason for this observation?

**Suitability:**

3

---

### Official Review · Reviewer_V7hX · 2024-05-26

**Rating:** 4
**Confidence:** 3

**Summary:**

The paper introduces SelM, a novel architecture for Audio-Visual Segmentation (AVS) tasks that addresses the limitations of current methods in dealing with auditory illusions. SelM employs a selective mechanism for noise reduction and robust feature selection, coupled with a dual alignment mechanism for early fusion of audio and visual features. It also includes a cross-level reasoning decoder to enhance segmentation precision. The method achieves state-of-the-art performance on various AVS benchmarks.

**Strengths:**

1. Novelty: The introduction of a selective mechanism for noise reduction and the use of dual alignment for early fusion in AVS tasks are innovative contributions.
2. Technical Correctness: The paper presents a well-structured methodology with sound theoretical underpinnings and a detailed explanation of the architecture.
3. Evaluation: The proposed method is thoroughly evaluated on multiple datasets and settings, demonstrating significant improvements over existing state-of-the-art methods.
4. Clarity: The paper is well-organized and clearly written, with comprehensive explanations of the proposed components and their integration.
5. Applications: The method has broad applicability in multimedia tasks, such as video editing, industrial maintenance, and surveillance, making it highly relevant to the field.

**Limitations:**

1. Complexity: The architecture introduces several new components which may increase the complexity and computational cost. Further discussion on computational efficiency and potential trade-offs would be beneficial.
2. Generalization: While the method shows strong performance on the evaluated datasets, additional experiments on more diverse and challenging datasets could further validate the robustness and generalizability of the approach.
3. Ablation Studies: Although the paper includes ablation studies, more detailed analyses of the individual contributions of each module and their interactions could provide deeper insights into the architecture’s performance.

**Suitability:**

3

---

### Meta-Review · Area_Chair_K8jw · 2024-06-23

**Recommendation:** Accept (Oral)
**Confidence:** 4

**Metareview:**

The authors address selective mechanism-based audio-visual segmentation. The proposed method is intriguing; however, the introduction of several new components could result in increased complexity and computational costs. It would be beneficial to provide more discussion on computational efficiency and potential trade-offs. While the method demonstrates strong performance on the evaluated datasets, additional experiments on more diverse and challenging datasets would better assess its generalizability. Moreover, analyzing the individual contributions of each module and their effects could be beneficial.